# Anti-Inflammatory and Anti-Bacterial Potential of Mulberry Leaf Extract on Oral Microorganisms

**DOI:** 10.3390/ijerph19094984

**Published:** 2022-04-20

**Authors:** Dokyeong Kim, Kyung-Hee Kang

**Affiliations:** 1Precision Medicine Research Center, College of Medicine, The Catholic University of Korea, Seoul 06591, Korea; dkkim2908@gmail.com; 2Department of Biomedicine & Health Sciences, College of Medicine, The Catholic University of Korea, Seoul 06591, Korea; 3Department of Dental Hygiene, Konyang University, Daejeon 35365, Korea

**Keywords:** mulberry leaf, anti-inflammatory agents, anti-bacterial agents, microorganisms, oral diseases

## Abstract

Mulberry leaves extract (*Morus alba* extracts; MAE) is known to have therapeutic potentials for numerous human diseases, including diabetes, neurological disorders, cardiovascular diseases, and cancers. However, there has not been sufficient research proving therapeutic effects on oral disease and its related oral risk factors. Thus, we investigated whether MAE has any anti-inflammatory and anti-bacterial effects on risk factors causing oral infectious diseases. To examine the anti-inflammatory response and bacterial inhibition of MAE, we measured intracellular reactive oxygen species (ROS) generation, production of pro-inflammatory cytokines, and the bacterial growth rate. Our study showed that MAE has anti-inflammatory activities, which inhibit the ROS generation and suppressed the production of pro-inflammatory cytokines (TNF-α and IL-6) in human monocyte THP-1 cells by stimulating lipopolysaccharide (LPS) and/or *F. nucleatum*, which are the virulent factors in periodontal diseases. Furthermore, MAE inhibited the bacterial growth on oral microorganisms (*F. nucleatum* and *S. mutans*) infected THP-1 cells. These findings suggested that MAE could be a potential natural source for therapeutic drugs in oral infectious disease.

## 1. Introduction

Over 700 species of microorganisms have been identified to exist in the oral cavity of healthy humans, and imbalance of host–microbial homeostasis can cause various oral diseases such as gingivitis, periodontitis, caries, and endodontic infections, and, in addition, further infectious diseases in various distant organ sites [1]. Among them, *Fusobacteria nucleatum* is a Gram-negative anaerobic bacillus and frequently isolated from both supra- and sub-gingival dental plaque [2]. It is pivotal in facilitating the structural bridge role in developing dental plaque and periodontal diseases, coaggregating with primary colonizers, including *Streptococcus oralis*, *Streptococcus sanguinis*, and *Streptococcus mitis*, etc., and anaerobic secondary colonizers (periodontitis-related keystone species) such as *Porphorymonas gingivalis* and *Aggregatibacter actinomycetemcomitans* [2,3]. Additionally, *Streptococcus* species are predominant in all oral niches. Out of them, *S. mutans*, a Gram-positive oral bacterium, has long been known to be the main pathogen of dental caries [4,5]. 

Natural resources have been used to treat various diseases and enhance human health. Mulberry (*Morus alba*), which belongs to family Moraceae, is multipurpose agroforestry plant that is native to China and widely distributed in many countries, including Asia, India, Africa, America, and Europe [6]. Mulberry leaves extract (*Morus alba* extracts; MAE) has various functions such as antitumor, antidiabetic, and anti-inflammatory activities, due to its bioactive compound that is rich in polysaccharides, polyphenols, alkaloids, and flavonoids [7,8]. Among its anti-microbial functions, its effect on some oral microorganisms such as *S. mutans* and *P. gingivalis,* has been reported in 2008 and 2015, respectively [9,10]. However, it is insufficient to explain anti-inflammatory and anti-microbial effects of MAE on oral microorganisms. Thus, we attempted to identify anti-bacterial and anti-inflammatory potentials of MAE on oral microorganisms (*F. nucleatum* and *S. mutans*), which are typical periodontitis and dental caries-related microorganisms, respectively. Additionally, we also examined the anti-inflammatory potentials of MAE in oral keratinocytes and fibroblasts. 

## 2. Materials and Methods

### 2.1. Preparation of Extracts

The mulberry leaf power (100%) was used for the research material and commercially purchased from Handsherb company (Yeoncheon-si, Korea). The mulberry leaf powder (60 g) and ethanol (600 mL) were mixed and left alone for 24 h at room temperature. Then, only the liquid component excluding precipitates was made to pass through a vacuum filter. Then, we concentrated the filtered extract using a decompression concentrator. The concentrated extract was stored to be analyzed after lyophilization.

### 2.2. Used Microorganisms and Culture

*Streptococuus mutans* KCTC 3065 was acquired from the Korean Collection for Type Culture (KCTC) and cultured in the Brain Heart Infusion (BHI) liquid media at 37 °C. *Fusobacterium nucleatum* ATCC 25586 was purchased from American Type Culture Collection (Manassas, VA, USA) and cultured in the BHI liquid media at 37 °C under the anaerobic condition. 

Optical density (OD) of bacteria cultures was measured using the plate reader (BioTek Instruments, Inc., Winooski, VT, USA), with 0.5 and 0.6 OD representing ~10^9^ CFU/mL for *S. mutans* and *F. nucleatum*, respectively.

### 2.3. Cell Culture

THP-1 cells, human monocytic cell lines, were acquired from the Korea Cell Bank (Seoul, Korea) and cultured in RPMI 1640 media with 10% fetal bovine serum (FBS), 100 U/mL penicillin, and 0.1 mg/mL streptomycin at 37 °C maintaining 5% CO_2_ condition. The tests were performed after seeding the cells onto 48-well plates (2 × 10^4^ cells/well) or 6-well plates (2 × 10^5^ cells/well). 

Immortalized human oral keratinocytes (IHOKs) [11] and gingival fibroblast (hTERT-hNOFs) [12] were provided by the department of the Oral Pathology in Yonsei University College of Dentistry. IHOK cells were immortalized by HPV16 E6/E7, and hTERT-hNOFs were immortalized by transfection with hTERT to be used in vitro. Both IHOKs and hTERT-hNOFs cells were cultured in DMEM/F12 media (3:1 ratio) with 10% FBS and 1% penicillin/streptomycin at 37 °C maintaining 5% CO_2_ condition.

### 2.4. Cell Viability Test 

The effect of MAE on THP-1 cell viability (>80% confluence of plates) was determined using a colorimetric EZ-CyTox Kit (Deaillab Service Co., Seoul, Korea) according to the manufacturer’s instructions. The absorbance was measured at 450 nm using a plate reader (BioTek Instruments, Inc., Winooski, VT, USA). 

IHOKs and hTERT-hNOFs were seeded onto 96-well plates by 1 × 10^4^ and treated with MAE with serum-free media (DMEM/F12 media (3:1 ratio)) for 24 h. Cell viability was measured by MTT (Duchefa, Haarlem, The Netherlands) assay. In brief, MTT solution was added and incubated at 37 °C for 4 h. After incubation, DMSO (MTT solvent) was added into each cell. The absorbance was read at 590 nm by using a microplate reader. All experiments were performed in triplicate.

### 2.5. Flow Cytometry

To check the generation of reactive oxygen species (ROS), THP-1 cells were seeded onto 6-well plates (2 × 10^5^ cells/well). The cells were pretreated with MAE by changing concentration levels for 2 h and stimulated by LPS (InvivoGen, San Diego, CA, USA) for 24 h. The stimulated cells were washed with PBS and dyed with DCFDA (Abcam, Cambridge, MA, USA) for 30 min. Living cells (10,000 cells) were screened based on their forward- and side-scatter profiles. The samples were collected using BD FACSCalibur Flow Cytometer (BD Biosciences, San Jose, CA, USA) and the data were analyzed by BD CellQuest Pro Software (BD Biosciences, San Jose, CA, USA). 

IHOKs (5 × 10^5^ cells/well) and hTERT-hNOFs (4 × 10^5^ cells/well) were seeded in the 6-well plate and pre-treated with the MAE using different concentrations (5, 10, and 20 mg/mL) for 1 h, and applied with 10 ng/mL LPS for 24 h. Next, fluorescent probe 2′7′-dichlorofluorescin diacetate (H_2_DCFDA) dye (Molecular Probes, Eugene, OR, USA) was used. According to instruction, both cells were applied with 10 μM of H_2_DCFDA dye in the dark at 37 °C for 20 min. ROS were analyzed by flow cytometry (Becton Dickinson, Beckman coulter).

### 2.6. Enzyme-Linked Immunosorbent Assay (ELISA)

THP-1 cells were pretreated with MAE for 2 h and then applied with LPS and *F. nucleatum* for 24 h. Cell culture supernatants were collected, and IL-6 and TNF-α were measured using the commercial ELISA kit (R&D Systems, Minneapolis, MN, USA). 

### 2.7. Bacterial Growth 

MAE (20, 40, and 80 mg/mL) was added to the BHI liquid media in the test group while no MAE was added in the control group. *S. mutans* and *F. nucleatum* (1 × 10^7^ bacteria) were inoculated and cultured in each media, then absorbance was measured using the optical density. Mean values were calculated by testing three times to gain the repeatable results in all test groups.

### 2.8. Statistical Analysis

The differences among the mean values of different groups were assessed. Data is expressed as the mean ± standard deviation. Statistical significance tests were performed using one-way analysis of variance followed by the Tukey post-hoc test using GraphPad Prism version 5.00 (GraphPad Software, Inc., La Jolla, CA, USA). *p* < 0.05 was determined to indicate a statistical significance.

## 3. Results

### 3.1. MAE Decreased the Production of ROS in LPS-Stimulated THP-1 and Oral Cells

To investigate the effects of MAE on cell viability, WST analysis was performed in THP-1 after treatment of MAE by various concentrations (10–80 mg/mL). Cells did not show significant cytotoxicity in the range of 10–40 mg/mL of MAE while the cell viability declined when the cells were treated with 80 mg/mL of MAE (Figure 1). 

Hence, MAE was used with 10, 20, and 40 mg/mL which proved to be non-cytotoxic. Next, we performed flow cytometry analysis to investigate whether MAE inhibits the increase in ROS by treatment with lipopolysaccharide (LPS), which is the richest component in the cell wall of Gram-negative germs. Results showed that LPS-induced ROS production was decreased in a dose-dependent manner when cells were pretreated with 10–40 mg/mL of MAE (Figure 2). 

In addition, we examined the effects of MAE in inflamed oral cells. Oral keratinocytes (IHOK) and fibroblasts (hTERT-hNOFs) were used to measure ROS generation in oral cells with MAE treatment. First, the MTT assay was performed to check the cytotoxicity of MAE. The cell viability was significantly reduced in IHOK with the treatment of 5, 10, 20, and 40 mg/mL of MAE. Meanwhile, the cell viability of hTERT-hNOFs declined significantly only when the cells were treated with 40 mg/mL of MAE (Appendix A). To perform the following test, the concentration levels of MAE were selected showing weak (within 80–60% of cell viability) or non-cytotoxicity (>80% of cell viability) in IHOKs and hTERT-hNOFs, respectively. Although ROS increased due to the inflammatory effect from LPS induction, MAE induced a decline in ROS by 3.8–4.8 times in LPS-stimulated IHOKs (Appendix A) and by 1.3–2.5 times in LPS-stimulated hTERT-hNOFs (Appendix A). 

Collectively, MAE could reduce ROS generation increased by LPS, indicating that MAE functions as an antioxidant in oral cells as well as monocytes. 

### 3.2. MAE Decreased the Generation of IL-6 and TNF-α in LPS/F. nucleatum-Stimulated THP-1 Cells

Next, we investigated whether MAE regulated production of inflammatory cytokines. First, we identified the effect of MAE in LPS-stimulated THP-1 cells. LPS induces the production of IL-6 and TNF-α, and both cytokines were markedly decreased after pretreatment with MAE in dose-dependent manner (Figure 3A,B). Next, we examined the effect of MAE in THP-1 infected by *F. nucleatum,* which is a typical germ involved in developing periodontal diseases. Infection of *F. nucleatum* led to generation of IL-6 and TNF-α and they were inhibited by pretreatment with MAE significantly (Figure 3C,D). 

Collectively, MAE could effectively reduce generation of pro-inflammatory cytokines in THP-1 cells stimulated by LPS and *F. nucleatum*.

### 3.3. MAE Has Antibiotic Effects against F. nucleatum and S. mutans

Last, we investigated whether MAE showed the antibiotic effects against *F. nucleatum* and *S. mutans,* which is known to be a pathogen causing periodontal diseases and dental caries, respectively. After completing culture of *F. nucleatum* in BHI liquid media with different concentration levels of MAE, absorbance was measured using the plate reader. We confirmed that bacterial growth was significantly declined in *F. nucleatum* with MAE more than *F. nucleatum* without MAE (Figure 4). Consistently, the growth of *S. mutans* declined in a dose-dependent manner of MAE. Therefore, anti-bacterial effects of MAE were confirmed against oral bacteria, such as *F. nucleatum* and *S. mutans.*

## 4. Discussion

Natural sources have been known to work as complementary and alternative medicines for centuries, with low levels of toxicity, therapeutic properties, and effects [13,14]. In contrast, some natural sources well known in traditional medicine have demonstrated that they have various toxicities [15]. However, *Morus alba* called mulberry has been evaluated as a safe herb showing no side effects [16,17]. The fruits, leaves, branches, and roots of mulberry possess a range of biological treatment effects on numerous human diseases [18,19], including diabetes, neurological disorders, cardiovascular diseases, and cancers [20,21,22,23,24]. However, there has not been sufficient research proving that mulberry can function as a therapeutic agent in oral diseases and its related oral risk factors. 

Inflammation is the process of recognizing and eliminating various harmful stimuli and initiating the healing process [25]. Pathogen infection can induce an inflammatory response through activation of pattern-recognition receptors (PRRs) [25,26]. The identification of PRRs by pathogens is mediated by various tissue-resident immune cells, leading to the release of inflammatory mediators, including chemokines and cytokines [27,28]. In addition, the antioxidant defense system affects oxidative stress. Increased oxidative stress can induce production of reactive oxygen species (ROS), which activate various transcription factors, including NF-κB, AP-1, p53, and STAT, followed by increased growth factors, inflammatory cytokines, and chemokines [29]. Based on such a background, we investigated inflammatory responses such as cytokines and ROS to identify the anti-inflammatory and anti-bacterial potentials of MAE. 

Monocytes and macrophages are critical cell types which recognize foreign pathogens using PRRs, and they secrete various pro-inflammatory chemokines and cytokines [30,31]. Thus, we mainly used THP-1 cell lines, which have been well known as human monocytes and macrophages in studies of inflammatory disease in vitro [31]. We applied LPS to induce inflammation in THP-1 cells because a key player in general bacteria-derived inflammation is LPS, also referred to as endotoxin [32,33]. First, we observed the effect of MAE as antioxidants in LPS-stimulated cells. MAE inhibited the increase in intracellular ROS production in LPS-stimulated THP-1 cells. Some researchers have consistently reported that MAE inhibits oxidative stress [34,35]. It means that MAE functions as an antioxidant in various cell types. However, there are few studies proving antioxidant effects of MAE in oral cells; there have been some studies showing anti-inflammatory effects in periodontal ligament cells (PDLs) [36,37]. Given that, we additionally examined ROS generation in oral cells. We selected two types of oral cells composed of oral mucosa; oral keratinocytes (immortalized human oral keratinocytes; IHOKs) [11] and fibroblasts (hTERT-transfected human normal oral fibroblasts; hTERT-hNOFs) [12]. Both cells were immortalized cell lines to use in vitro by transfecting HPV16 E6/E7 and hTERT, respectively. It was shown that LPS-stimulated IHOK and hTERT-hNOFs increased intracellular ROS, and then it was decreased by MAE. Thus, our findings suggested that MAE could function as antioxidants in oral inflammation. 

Periodontitis and caries are the major oral diseases, in relation to oral microbiota [38]. It is now well-established that *S. mutans* are associated with initiation of dental caries by acid production, sugar fermentation, and acid tolerance [4,39]. Periodontitis is an infectious and inflammatory disease characterized by progressive infiltration of bacteria and inflammatory cytokines into periodontal tissues, resulting in attachment loss, alveolar bone destruction, and the loss of teeth [40,41]. Among prime periodontal disease-related bacteria, *P. gingivalis*, Gram-negative, and anaerobic bacteria, are one of major causative pathogens which are found in > 85% of subgingival plaque of periodontitis patients [42]. Especially, *P. gingivalis*-LPS is a strong virulence factor in periodontitis and induces cytokine secretion (TNF-α, IL-6, and MCP-1) and inflammatory responses via TLRs [43]. The results of our study are consistent to various other research showing anti-inflammatory effects of MAE in *P. gingivalis*-infected THP-1 cells [44]. In addition to *P. gingivalis*, *F. nucleatum* is another key-stone periodontal pathogen. In dental plaque, *F. nucleatum* plays a structurally supportive and interconnecting role between commensal early colonizers and more pathogenic late colonizers, indicating that *F. nucleatum* acts as a backbone in progression of periodontal diseases [2]. Hence, we focused on *F. nucleatum* as the main causative bacteria of periodontitis more than *P. gingivalis.* To mimic the inflammatory response in periodontal tissues, we applied with LPS and *F. nucleatum*, the main causative factors in periodontal diseases. Our study showed that LPS-induced TNF-α and IL-6 were reduced by MAE in THP-1 cells. Additionally, *F. nucleatum*-infected THP-1 cells showed the increase in TNF-α and IL-6 and they were downregulated by MAE. Collectively, MAE was found to block the secretion of pro-inflammatory cytokines increased by stimulation of various periodontal pathogens, indicating that MAE might have potential for developing therapeutic herbal medicines in periodontal diseases. Even *F. nucleatum* can have a pathogenic role in extra-oral diseases such as colorectal cancer, inflammatory bowel disease, and rheumatoid arthritis [45,46,47]. Thus, MAE might be effective herbs in various human diseases caused by *F. nucleatum*, as well as periodontal disease. Lastly, we observed the effect of MAE in inhibiting bacterial growth. We selected two typical oral disease-related bacteria (*S. mutans* and *F. nucleatum*) and found that the growth of both bacteria was inhibited by MAE, varying in different doses. Our results are consistent with other studies showing that MAE have anti-inflammatory and anti-bacterial effects on oral bacteria. For example, the component (1-deoxynojirimycin (DNJ)) of MAE showed anti-adhesive effects by controlling the overgrowth and inhibiting biofilm formation of *S. mutans* [9]. Additionally, MAE have anti-inflammatory effects by suppressing MMPs, tissue destruction-related protein, in *P. gingivalis* LPS-infected THP-1 cells [44]. Taken together, MAE could function as anti-bacterial and anti-inflammatory agents in oral microorganisms.

We demonstrated the anti-inflammatory and anti-bacterial effects of MAE on risk factors causing oral infectious diseases. However, there are some limitations of this study. THP-1 cells were used to investigate the anti-inflammatory response of MAE, while oral cells were only used to identify the antioxidant effects. However, one study has verified that LPS increases inflammatory cytokines such as IL-1β, IL-6, and IL-8 in hTERT-hNOFs and IHOKs which our study used [48]. Additionally, we need to further study whether *F. nucleatum* can increase cytokines, similar to results by LPS. Additionally, we did not observe the anti-inflammatory and anti-oxidant mechanisms of MAE in THP-1 cells, but only observed the anti-inflammatory phenomenon. Our findings are consistent with some other studies showing that IL-6 and TNF-α were increased by LPS in macrophages [49,50]. The inflammatory response can be orchestrated by pro-inflammatory cytokines such as TNF-α and IL-6 [25,51,52] and it can be mediated through NF-κB transcriptional factor [44,52]. Additionally, ROS can be regulated by NF-kB [53]. Based on other studies, we can speculate that anti-inflammatory signals of MAE might be involved in NF-kB-mediated signaling pathways [41]. Additionally, we need to identify what components of MAE can trigger anti-inflammatory and anti-bacterial effects. Phytochemical studies have revealed that MAE have many bioactive constituents, including flavonoids, cumarins, phenolic acid, alkaloids, and terpenoids [8]. In particular, flavonoids and phenolic acid in MAE have functions as antioxidants and anti-bacterial effects [18,19]. Considering that, we also speculate that flavonoids and/or phenolic acid might inhibit the ROS generation and production of pro-inflammatory cytokines in MAE. To solve these limitations, further studies are needed to clarify what components of MAE are considered in anti-inflammatory functions and their mechanism in detail. 

## 5. Conclusions

In conclusion, we demonstrated that MAE has anti-inflammatory effects, which can inhibit ROS generation and suppress production of pro-inflammatory cytokines in human monocyte THP-1 cells by stimulating LPS and/or *F. nucleatum*. Furthermore, MAE can inhibit the bacterial growth in oral microbes *F. nucleatum* and *S. mutans*. These findings suggested that MAE could be a potential natural source for prophylactic and therapeutic agents in oral infectious disease. 

## Figures and Tables

**Figure 1 ijerph-19-04984-f001:**
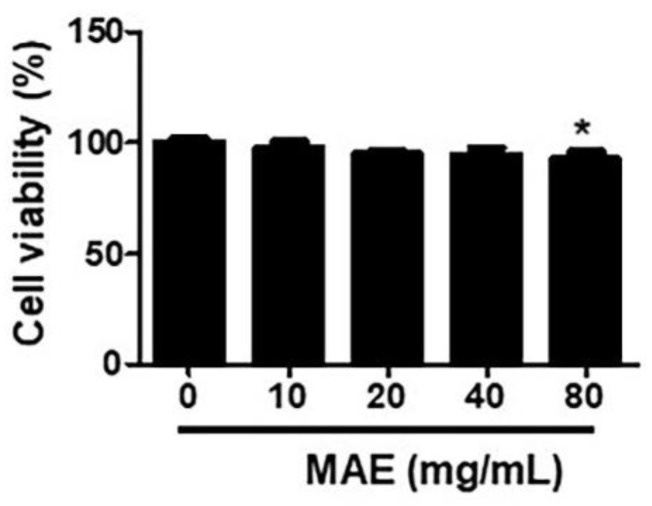
**The cytotoxicity test of mulberry leaves extract (*Morus alba* extracts; MAE) in THP****-1 cells.** The cell viability of THP-1 in various concentration levels of MAE. THP-1 was treated with 10, 20, 40, and 80 mg/mL of MAE for 24 h. Data were analyzed by one-way ANOVA. * *p* < 0.05.

**Figure 2 ijerph-19-04984-f002:**
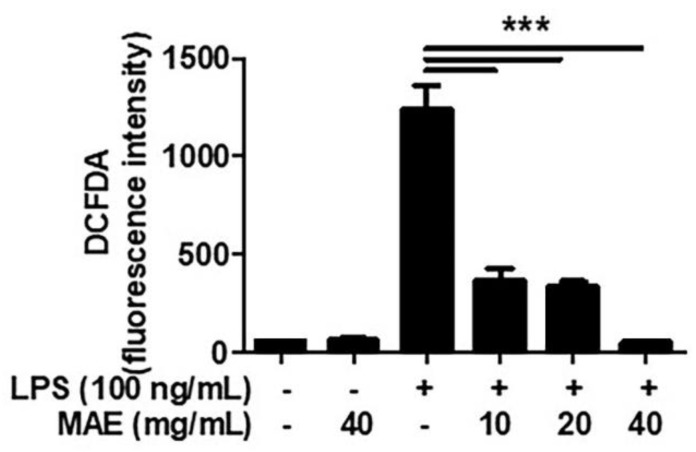
**MAE reduced generation of reactive oxygen species (ROS) in THP****-1 cells stimulated by lipopolysaccharide (LPS).** THP-1 was pretreated with MAE for 2 h and stimulated by LPS (100 ng/mL) for 24 h. ROS expression in the cell was shown by flow cytometry using a DCFDA antibody. The data were analyzed using one-way ANOVA. *** *p* < 0.001.

**Figure 3 ijerph-19-04984-f003:**
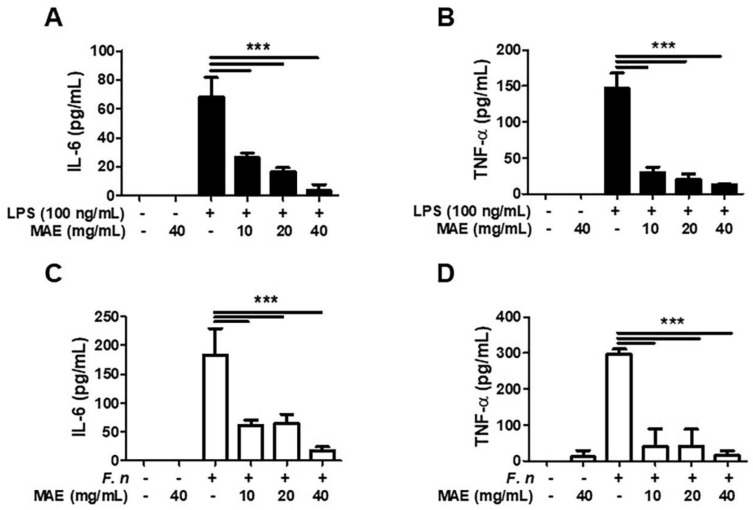
**The effect of MAE inhibiting generation of cytokines in THP-1 cells stimulated by LPS/*F. nucleatum*.** THP-1 was pretreated with different concentrations of MAE for 2 h and stimulated by either LPS (100 ng/mL) or *F. nucleatum* (MOI 10) for 24 h. The concentrations of IL-6 (**A**,**C**) and TNF-α (**B**,**D**) were measured by ELISA in the culture supernatant. The data were analyzed using one-way ANOVA. *** *p* < 0.001.

**Figure 4 ijerph-19-04984-f004:**
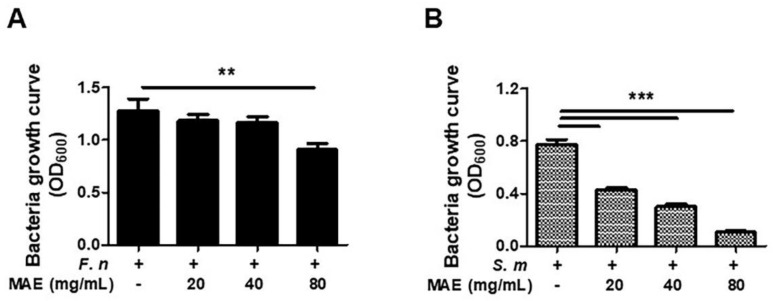
**The effect of MAE in inhibiting growth of both *F. nucleatum* and *S. mutans*.** MAE (0, 20, 40, and 80 mg/mL) was added in BHI liquid media, *F. nucleatem* (**A**) was inoculated (1 × 10^7^ bacteria), and it was cultured for 24 h. *S. mutans* (**B**) was inoculated (1 × 10^7^ bacteria) and it was cultured for 12 h. The absorbance was measured by optical density. The data were analyzed using a one-way ANOVA. ** *p* < 0.01, *** *p* < 0.001.

## Data Availability

All data are available on request from the corresponding author.

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
