# Peer review of "Anti-Inflammatory and Anti-Bacterial Potential of Mulberry Leaf Extract on Oral Microorganisms"

_ijerph, 2022, doi:10.3390/ijerph19094984_

Round 1
Reviewer 1 Report
Dear Editor of the International Journal of Environmental Research and Public Health,
I reviewed the manuscript ijerph-1651931entitled “Anti-inflammatory and anti-bacterial potentials of mulberry leaf extract on oral microorganisms” submitted by Dokyeong Kim et al. The authors present a concise work for the investigation of the properties of the mulberry extract on bacteria of periodontal disease. They found that the extract diminished the ROS production and pro-inflammatory cytokines on THP1 cells. Although the mechanism of the MAE extract is not presented in the study, the authors recognized the importance of further studies to elucidate the anti-inflammatory function of MAE extract. I consider the manuscript to be well written and presented. The methods and results correlate, and the discussion is well supported. Therefore, I recommend the publication of the manuscript. However, I have a few comments to improve the work:
• Indicate the source of the mulberry leaf powder.
• In figure 1, indicate whether the percentage of MAE is related to mass or volume. For example, is it possible to show the amount of MAE in mass per mL?
Author Response
We have reviewed the entirely, and English proofreading was also performed (corrected sentences or words corrected by English editing service are marked in red color).
We attached file about a point-by-point response to the reviewer's comments.

Reviewer 2 Report
„the reports are not enough that mulberry functions as therapeutic agent in oral disease and its related oral pathogen“ – there isn't a sufficient number of reports confirming the effect of mulberry leaf extract in oral diseases and on the associated pathogens.
„pathogens (LPS, F. nucleatum and S. mutans) of oral infectious diseases“ – pathogens causing oral infectious diseases
„To examine the inflammatory response of MAE, we measured the intracellular ROS generation, the production of pro-inflammatory cytokines, and the bacterial growth rate“ - inflammatory response caused by MAE. In the original sentence it sounds like the inflammatory response is made by the extract and not the oral tissue. Also, measuring the bacterial growth rate is quantifying the antibacterial effect, it does not measure the anti-inflammatory response.
„by stimulation of LPS or/and F. nucleatum, which are the virulence pathogens“ – by stimulation with LPS. In the original sentence it sounds like the LPS was stimulated. There are no virulence pathogens, there are virulent pathogens.
„anti-inflammatory and anti-bacterial effect of MAE on pathogens (LPS, F. nucleatum and S. mutans)“ While LPS ia a class of bacterial products, LPS itself is not a pathogen. As a possible cause of a disease, it represents a toxin.
Some sentences are simply nonsense – „MAE inhibited the bacterial growth inhibition in oral microorganisms“ Inhibited the growth inhibition?!
„various oral disease“ - various oral diseases.
„the imbalance of host-microbial homeostasis can cause various oral disease such as gingivitis, periodontitis, caries, and endodontic infection, and further infectious diseases in digestive and respiratory tract“ – endodontic infections. There are various causing agents of different endodontic infections. Also, why limit further infectious diseases to digestive and respiratory tracts only?
„Fusobacteria nucleatum is gram-negative anaerobic bacilli“ - Fusobacteria nucleatum is a Gram-negative anaerobic bacillus. et al. is an abbreviation of et alia and is written with a dot. „Streptococcus species is predominant in all oral niches.“ Streptococcus species are predominant in all oral niches.
„Mulberry (Morus alba), which belongs to family Moraceae, is multipurpose agro-forestry plant that is widely distributed in China, India, Middle East, Southern Europe, and recently in North America“ There isn't just one mulberry species. Morus alba, white or common mulberry, is only one of at least around twenty species, not to mention the plants in the related genus Broussonetia that are also reffered to as „mulberry“. Recently?! Widely distributed? It was introduced to North America four centuries ago. It is distributed and became naturalized worldwide. They started the silk industry in Brasil in the 18th century, now Brasil is the world's fifth largest producer of silk yarn, yet authors don't even mention the South America.
„little has been known about its anti-inflammatory effect and anti-microbial properties in F. nucleatum and inflamed oral cells.“ once again, the authors fail to clearly distinguish antibacterial and anti-inflammatory effects. Not „in F. nucleatum“, on F. nucleatum.
„Optical density (OD) was measured for the bacteria using the plate reader (BioTek Instruments, Inc., Winooski, VT, USA), representing 0.5 and 0.6 (~ 109 CFU/mL) as S. mutans and F. nucleatum, respectively“ - Optical density (OD) of bacterial cultures was measured using the plate reader (BioTek Instruments, Inc., Winooski, VT, USA), with 0.5 and 0.6 representing ~ 109 CFU/mL for S. mutans and F. nucleatum, respectively.
- Materials and methods mention only THP-1 cell line. Yet, 3.1 in the results suddenly introduces experiments with TERT-hNOF and IHOK cell lines. The same applies to WST analysis. Supplementary Materials and Methods are short – I fail to see why those data should be supplementary and not mentioned in the Materials and methods section of the manuscript.
What does „TERT-hNOFs was not found by treatment with 0.5% to 2% MAE“ mean?!? I assume it is meant to say no reduced viability has been observed with TERT-hNOF cells. And if the cell viability was significantly reduced in IHOK by every MAE concentration („the cell viability was significantly declined in IHOK by 0.5%, 1%, 2% and 4% of the MAE“), how both cell lines could be used for determination of oxidative stress levels („the concentration to show over 80% of cell viability in both cell lines was selected to perform the following test“)?
What does „First, we identified the effect of MAE in LPS-stimulated THP-1 cells. As results, LPS-induced IL-6 and TNF-α were decreased by MAE“ mean? The authors analysed the effect of MAE in LPS-stimulated THP-1 cells. A decrease in production of IL-6 and TNF-α was observed after treatment with MAE.
Why was cytokine production analysed in THP-1 cells only? If the anti-inflammatory effect was searched for in „inflamed oral cells“ as stated in the introduction, why the authors omitted to analyse cytokine produced by oral keratinocytes and fibroblasts?! Both produce cytokines after stimulation with LPS.
The authors state „Natural products have been known as an important source for developing the complementary and alternative medicines for centuries, due to low level of toxicity, therapeutic properties, and efficacy“. While that might be true for some products, such generalizations are needless and dangerous. If the authors do not wish to indulge in scientific literature, a brief glance of news reports is enough to warn of „natural products“: eg. https://edition.cnn.com/2017/03/21/health/poisoned-herbal-tea-death-san-francisco/index.html
What does the sentence „Among many natural products, Morus alba called to Mulberry has found in many countries, especially in Asia“ mean?
„Inflammation is a pivotal immune response that maintain the balances in tissue homeostasis during infection of injury under a variety of noxious conditions“ What are „balances in tissue homeostasis“? And what are the „noxious conditions“ mentioned? What about infections that have nothing to do with injury? Or those that have no „noxious conditions“?
What does „LPS-induced inflammatory cytokines such as IL-6, and TNF-α showed increased in macrophage“ mean?
Author Response
We have tried our best to address the editor’s and reviewer's comments and suggestions.
We have reviewed the entirely, and English proofreading was also performed (corrected sentences or words corrected by English editing service are marked in red color).
We attached file about a point-by-point response to the reviewer's comments.
Best wishes and many thanks for this opportunity to submit our revised manuscript,
Sincerely,

Reviewer 3 Report
Dear Authors,
the manuscript is prepared in a clear and transparent manner. However, I am asking for absolute supplementation of the information regarding the subject of research:
- Growing conditions (this is important!): How many days did the cultivation take ?, what light / intensity / photoperiod? tempratura? how was growth monitored? what soil? where did the seeds / seedlings come from?
I must admit that, due to the subject matter, I will recommend that the text be transferred to another journal at the MDPI publishing house.
Author Response

(The authors gave the same response as above.)

Round 2
Reviewer 3 Report
Dear Authors,
thank you for the submitted manuscript and the corrections made.
Best regards,